# Comparative Meta-Analysis of Long-Read and Short-Read Sequencing for Metagenomic Profiling of the Lower Respiratory Tract Infections

**DOI:** 10.3390/microorganisms13102366

**Published:** 2025-10-15

**Authors:** Giovanni Lorenzin, Maddalena Carlin

**Affiliations:** 1U.O.M. Microbiology and Virology, APSS, S.Chiara Hospital, 38122 Trento, Italy; 2Department of Cellular, Computational and Integrative Biology (CIBIO), University of Trento (UNITN), 38123 Trento, Italy

**Keywords:** metagenomics, nanopore, illumina, long-read sequencing, short-read sequencing, lower respiratory tract infection

## Abstract

Metagenomic next-generation sequencing (mNGS) is increasingly employed for the diagnosis of lower respiratory tract infections (LRTIs). However, the relative diagnostic performance of long-read versus short-read sequencing platforms remains incompletely defined. For this systematic review, a search was conducted in PubMed, Embase, Scopus, Web of Science, and Google Scholar to identify studies directly comparing long-read (e.g., Oxford Nanopore, PacBio) and short-read (e.g., Illumina, Ion Torrent, BGISEQ) metagenomic sequencing for the diagnosis of LRTI. Eligible studies reported diagnostic accuracy or comparative performance between platforms. Risk of bias was evaluated using the QUADAS-2 tool. Thirteen studies met inclusion criteria. Reported platforms included Illumina, Oxford Nanopore, PacBio, Ion Torrent, and BGISEQ-500. A total of 13 studies met inclusion criteria. Across studies reporting sensitivity, average sensitivity was similar for Illumina (71.8%) and Nanopore (71.9%). Specificity varied substantially, ranging from 42.9 to 95% for Illumina and 28.6 to 100% for Nanopore. Concordance between platforms ranged from 56 to 100%. Illumina consistently produced superior genome coverage (approaching 100% in most reports) and higher per-base accuracy, whereas Nanopore demonstrated faster turnaround times (<24 h), greater flexibility in pathogen detection, and superior sensitivity for Mycobacterium species. Risk of bias was frequently high or unclear, particularly in patient selection (6 studies), index test interpretation (5), and flow and timing (4), limiting the robustness of pooled estimates. Long-read and short-read mNGS platforms exhibit comparable strengths in the diagnosis of LRTIs. Illumina remains optimal for applications requiring maximal accuracy and genome coverage, whereas Nanopore offers rapid, versatile pathogen detection, particularly for difficult-to-detect organisms such as Mycobacterium. However, there are certain limitations of the review, including a lack of comparable outcomes reported in all studies; therefore, further research is warranted to address this.

## 1. Introduction

Lower respiratory tract infections (LRTIs) remain a significant contributor to morbidity and mortality globally. In 2021, approximately 344 million people suffered from LRTIs, causing 2.18 million deaths [1]. LRTIs are time-sensitive conditions where delays or inaccuracies in pathogen identification can directly affect patient survival and antibiotic stewardship. Early and precise diagnosis not only enables targeted antimicrobial therapy but also reduces unnecessary broad-spectrum antibiotic use, which is a major driver of antimicrobial resistance [1,2]. Traditional culture-based diagnostics for LRTIs have well-known limitations. Only a subset of lung pathogens grow readily in laboratory media, and culture can be slow and biased by prior antibiotic exposure [3]. In contrast, culture-independent metagenomic sequencing can detect virtually all microbial DNA in a specimen at once [4]. Metagenomic next-generation sequencing (mNGS) has emerged as a powerful tool for infectious disease diagnosis, as it can identify bacteria, fungi, viruses, and other microbes simultaneously from respiratory samples [5]. Recent clinical studies confirm that mNGS improves pathogen detection in pneumonia. For example, an ICU study of pediatric pneumonia showed that mNGS had high sensitivity and detected fungi and mixed infections that conventional methods missed [5].

Metagenomic sequencing of clinical samples relies on high-throughput DNA sequencers. The two main approaches are short-read sequencing (e.g., Illumina platforms) and long-read sequencing (e.g., Oxford Nanopore Technologies [ONT] and Pacific Biosciences [PacBio]). Each offers distinct advantages for metagenomics. Short-read platforms like Illumina currently dominate microbiome research. These instruments produce large volumes of highly accurate reads, typically 75–300 base pairs in length per read [6,7]. The high accuracy (>99.9% per base [7]) and depth of coverage enable precise base-calling and reliable detection of variants. Paired-end Illumina reads afford robust coverage across genomes, which is ideal for single-nucleotide polymorphism (SNP) analysis and phylogenetics [7]. However, the short length of Illumina reads poses challenges. In microbial communities with many similar strains or repetitive elements, assemblies from 100 to 300 bp reads tend to be fragmented into hundreds of contigs. This fragmentation can hinder reconstruction of complete genomes and may limit strain-level resolution [8]. In practice, Illumina-based metagenomes often recover partial genomes but leave repeats and mobile elements unresolved [9]. Despite these limitations, Illumina sequencing remains relatively low-cost and high-throughput [10].

Long-read platforms address some of these limitations by generating much longer DNA reads. ONT’s MinION or GridION and PacBio’s Sequel II can routinely produce reads of several kilobases (often 5–20 kb or more) [11]. These long reads can span entire repeat regions, operons, and even whole small microbial genomes, greatly simplifying genome assembly. Studies show that long-read metagenomes yield more contiguous assemblies and higher recovery of complete MAGs than short reads [12]. Long reads also capture complete genes and operons intact, which improves functional annotation and detection of structural variants or antibiotic-resistance cassettes. However, long-read technologies have historically had higher per-base error rates than Illumina. Early ONT and PacBio reads had raw accuracies on the order of 90–95%, although recent chemistries (PacBio HiFi, ONT R10+) now approach 99% accuracy [12]. It is also important to note that long-read platforms vary in throughput and cost. For example, a single MinION run may yield 10–20 Gb of data [13], whereas high-end PacBio systems can output on the order of tens to hundreds of Gb.

Although some systematic reviews and meta-analyses have investigated the performance of long-read or short-read metagenomics [14,15], no systematic review has focused on the comparative performance of long-read versus short-read metagenomics in LRTIs. This leaves a gap in the literature. Therefore, this systematic review was conducted to fill this evidence gap and guide future clinical application of metagenomic sequencing in LRTIs.

## 2. Methods

This systematic review was carried out according to the Preferred Reporting Items for Systematic Reviews and Meta-Analyses (PRISMA) guidelines [16]. The review protocol was prospectively registered in the International Prospective Register of Systematic Reviews (PROSPERO; RN:CRD420251143392).

### 2.1. Search Strategy

For this systematic review, a comprehensive search was conducted in several key databases, including PubMed, Scopus, Embase, Web of Science, and Google Scholar, to identify potential studies investigating low-read versus short-read metagenomics in LRTI. The search was carried out using a combination of keywords including “Metagenomics,” “Illumina,” “short-read,” long-read,” and “lower respiratory tract infection”. These keywords were combined by AND and OR Boolean operators. Complete details of search strategy is presented in Appendix A.

### 2.2. Study Eligibility

The PICOS for the systematic review included population (P): patients suffering from LRTIs. Intervention (I): short-read metagenomics. Comparator (C): long-read metagenomic. Outcome (O): Diagnostic performance metrics (e.g., sensitivity, specificity, taxa detected) and microbiome profiling characteristics (e.g., richness, diversity, resolution at species level, pathogen detection capability); study design (S): randomized controlled trials, cohort studies, cross-sectional studies, and validation studies. Studies were included if they investigated low-read versus short-read metagenomics in LRTIs, irrespective of study year or study design. However, studies that solely focused on a single approach and did not report comparative outcomes for both approaches were excluded. Furthermore, studies that were published in any non-English language were also excluded.

### 2.3. Study Selection

After retrieving the results from the database search, the final files were exported to Rayyan (web version, accessed on 2 July 2025, available at: https://rayyan.ai), a software specifically designed for the systematic review screening process [17]. Before beginning the screening process, duplicates were detected and removed. Two independent reviewers (AB, PA) were involved in the screening process. Both reviewers were blinded to each other’s decisions. In the first step, records were screened based on titles and abstracts. After that, both reviewers compared their decisions and finalized the study selection based on full-length screening. In case of any disagreements, a third reviewer (LU) was involved. Data was extracted from each study, including study design, sample size, participant demographics, intervention details, and outcomes in an Excel sheet.

### 2.4. Data Synthesis

Data were extracted as reported in the original publications. No attempts were made to calculate or derive unreported outcomes such as sensitivity or specificity from supplementary files, text, or figures. If a study did not provide specific diagnostic performance values, those fields were marked as “not available” in the evidence tables. For consistency, diagnostic accuracy measures (sensitivity, specificity, concordance, and agreement rates) were presented as percentages. Turnaround times were converted to hours or days where necessary to allow comparisons. Read metrics and coverage were summarized using the reported units. Due to substantial heterogeneity in study design, reporting metrics, and outcome measures, a quantitative meta-analysis was not feasible. Instead, a structured narrative synthesis was conducted.

### 2.5. Quality Assessment

The quality of the included studies was assessed using the Quality Assessment of Diagnostic Accuracy Studies (QUADAS-2) tool [18]. Two independent reviewers (AB, PA) were involved in the quality assessment process. In case of any disagreements, a third reviewer (LU) was involved. QUADAS-2 measures two main domains—risk of bias and applicability concerns. Regarding risk of bias, four sub-domains are assessed, including patient selection, index test, reference standard, and flow and timing, whereas applicability concern focuses on three domains, including patient selection, index test, and reference standard.

### 2.6. Outcome Measures

The main outcomes assessed in this systematic review were sensitivity and specificity. The secondary outcomes measured were concordance/agreement (%), positivity rate, turnaround time, genome coverage (%), and read metrics. If any other outcomes were reported in studies, they were also reported.

## 3. Results

### 3.1. Included Studies

A total of 187 articles were identified from database searches, with 74 studies identified from PubMed, 2 from Web of Science, 54 from Scopus, 40 from Embase, and 17 studies from Google Scholar. After removing all the irrelevant studies, 13 studies were included in the systematic review.

### 3.2. Flow Diagram

Figure 1 shows the PRISMA flow diagram of this systematic review.

### 3.3. Study Characteristics

Table 1 shows the characteristics and diagnostic accuracy of included studies. Regarding study design, four were prospective, four were cross-sectional, whereas three studies were retrospective. One study used a methodological validation study design. The number of participants varied considerably, ranging from single-patient case reports [19] to studies analyzing over 100 samples [20]. Several studies examined clinical samples rather than individual patients. Age reporting was inconsistent; where available, participants were mostly middle-aged to elderly, with median ages typically between 55 and 70 years. Gender distribution was poorly documented, with only two studies reporting that approximately 63% of participants were male [21,22]. The majority of the studies (*n* = 9) compared Nanopore with Illumina [21,23]. Capraru et al. [20] compared Ion torrent with Nanopore, whereas Carbo et al. [24] compared Illumina, Ion Torrent, and Nanopore. Similarly, Wang et al. [19] compared Nanopore with BGISEQ-500 and Hahn et al. [25] compared PacBio with Illumina. Sensitivity was reported in five studies, with three studies reporting sensitivity for both Illumina and Nanopore. Among these three studies, average sensitivity for Illumina was 71.8%, whereas for Nanopore, it was 71.9%. Specificity was reported in only four studies; Illumina values ranged from 42.9% to 95%, while Nanopore values ranged from 28.6% to 100% where available. Overall, specificities for long reads vs. short reads were comparable.

Table 2 shows performance characteristics and outcomes of various platforms. Agreement between platforms and with diagnosis varied greatly, ranging from 56% to 100%. Several studies reported perfect or near-perfect concordance, such as Wang et al. [19], Lewandowski et al. [28]. Positivity rates were inconsistently reported. Where available, Illumina showed slightly higher positivity for certain pathogens: Illumina 91% vs. Nanopore 78% [27]); however, this trend was not consistent, as shown by Ma et al. [22]. Turnaround time was reported by only a few studies but showed lower turnaround time in Nanopore compared to Illumina. Genome coverage percentages were variable. Illumina generally achieved high coverage (close to 100% in studies), but Nanopore often achieved comparable or complete coverage when sequencing depth thresholds were met (Li et al. 2020 reported ≥ 97.6% completeness [23]). Illumina typically produced higher total read counts, while Nanopore generated fewer reads but with deeper per-read coverage (Carbo et al. 2023: depth > 2000 [24]). Across studies, Nanopore consistently offered faster turnaround and broader pathogen detection. Illumina, however, maintained higher accuracy in genome coverage, read quality, and stability.

### 3.4. Methods Quality Assessment

Figure 2 and Figure 3 show the risk of bias and applicability concerns of the included studies. Regarding risk of bias, high risk of bias was observed in three studies in the patient selection domain, whereas three studies had unclear risk of bias. Applicability concerns regarding patient selection were also high risk in four studies, whereas two studies had unclear risk. Overall, almost half of the studies had a low risk of bias in patient selection and applicability concerns. In the index test, three studies had a high risk of bias, whereas two studies had an unclear risk of bias. Applicability concerns also had high risk in four studies, whereas two studies had unclear risk in the index test domain. Overall, the risk of bias was low regarding the index test in the majority of the studies. Reference standard domain also showed a high risk of bias in two studies and an unclear risk of bias in two studies. However, six studies had high risk concerns regarding applicability in this domain, with one study having unclear risk. Flow and timing had a high risk of bias in four studies, with two studies having an unclear risk of bias.

## 4. Discussion

This systematic review, based on evidence from 13 studies, compared long-read (primarily Nanopore) and short-read (Illumina, Ion Torrent, BGISEQ) metagenomic sequencing for profiling LRTIs. To the best of our knowledge, this is the first systematic review that has compared long-read versus short-read metagenomic sequencing for diagnosing LRTIs. The findings of our systematic review showed that both approaches have comparable capabilities in detecting pathogens but with distinct trade-offs. Long-read platforms like Nanopore consistently offered faster turnaround times and broader taxonomic coverage, whereas short-read Illumina sequencing generally produced higher data quality and coverage but required longer processing times [32]. Our findings align with previous published evidence that shows that Nanopore offers faster turnaround compared to Illumina [33]. Several factors can be attributed to the faster turnaround time by Nanopore. For example, library preparation in the Nanopore approach is straightforward. It allows direct reading without reverse transcription or amplification, avoiding multiple steps required by Illumina. This streamlined process saves time and reduces errors [34]. For samples with low viral concentrations, Nanopore platforms can integrate various amplification methods to enhance accuracy. With a rapid barcoding kit, building an amplified sublibrary may take only ten minutes. Secondly, Nanopore devices provide information continuously as nucleic acids travel through the pore, which enables immediate data generation [35].

Although there is a paucity of data that has compared short-read and long-read metagenomics, several previous studies support our findings. For example, researchers found that Illumina and Nanopore had similar sensitivity for fungal detection (91% each), but Nanopore achieved perfect specificity (100%) versus 89% for Illumina. Conversely, Illumina was slightly more sensitive for bacteria (79% vs. 75%) [36]. Similarly, a systematic review and meta-analysis by Guo et al. investigated the effectiveness of mNGS for LRTIs. Across bronchoalveolar lavage samples, mNGS showed a pooled sensitivity of 89% and specificity of 90% [15]. Although mNGS was very effective for confirming LRTI pathogens, they did not compare long-read with short-read. In the present systematic review, some studies also reported that Nanopore detected more diverse taxa in respiratory samples than Illumina. For instance, Zhang et al. reported that Nanopore detected more viruses, fungi, and *Mycobacterium* compared to Illumina; however, results were comparable for bacteria [21]. Similarly, Ma et al. reported that Nanopore sequenced 90.9% of *Mycobacterium* infections versus only 36.4% by Illumina [22]. However, Heikema et al. reported that Nanopore is not that effective with genus Corynebacterium [27].

In contrast, Illumina platforms consistently generated higher sequencing depth and genome coverage when targets were present. Illumina runs typically produce far more total reads per sample [7]. This depth translates to excellent coverage of microbial genomes. Carbo et al. [24] reported Illumina coverage of 99.8% of the SARS-CoV-2 genome, compared to 81.2% by Nanopore. By contrast, Nanopore sometimes achieved complete assemblies at high depth. Li et al. [23] found ≥97.6% SARS-CoV-2 genome completeness by Nanopore at ≥250× depth. In practice, Illumina’s uniform short reads offer very high base-call accuracy, which makes it more reliable for fine-scale SNP calling and low-abundance variants, whereas Nanopore’s higher per-read error can reduce base accuracy but can still provide near-complete genomes when many long reads overlap [7]. However, Luan et al., in their study, showed that while long-read polishing alone improved assemblies, combining both long- and short-read polishing was necessary to achieve near-perfect accuracy [37].

In our systematic review, sensitivity and specificity were comparable between two approaches. Yan et al., in their meta-analysis, showed that clinical mNGS, regardless of platform, improves pathogen detection and can impact treatment decisions [14]. Our results complement these by detailing platform-specific trade-offs. Individual studies outside LRTI have similarly noted that Oxford Nanopore provides rapid diagnosis and comparable sensitivity to Illumina in diverse body fluids [36]. However, some authors caution that Nanopore’s higher base–error rate may limit species resolution without careful calibration. Similarly, bench comparisons in environmental DNA have shown that Illumina remains more efficient for species detection when DNA is degraded [38]. Overall, our conclusions are consistent with the emerging consensus that long- and short-read mNGS are complementary, with long reads for speed and breadth and short reads for accuracy and depth. Despite the promising diagnostic performance of both long-read and short-read mNGS observed in this review, several practical challenges limit their routine clinical adoption. First, laboratory infrastructure and cost remain important barriers. Long-read platforms such as Nanopore demand ongoing consumable expenditure and frequent flow cell replacement, while high-throughput short-read instruments require substantial capital investment and centralized facilities [39]. Second, sample handling and low microbial biomass in many respiratory specimens make host nucleic acid depletion and contamination control critical; procedural variability between laboratories can markedly influence sensitivity and false-positive rates [39].

LRTIs present several unique challenges for metagenomic sequencing compared with other infection sites, and these differences can influence the relative performance of long- versus short-read platforms. Respiratory samples such as sputum, bronchoalveolar lavage fluid (BALF), or tracheal aspirates are often highly heterogeneous, containing variable proportions of host nucleic acids, commensal flora, and potential pathogens, which complicates accurate signal-to-noise discrimination [40]. In addition, many LRTIs involve mixed infections or co-pathogens (e.g., viral–bacterial or bacterial–fungal coinfections). Platform-specific strengths and limitations also become important in this context. Long-read approaches like Nanopore may better resolve complex or repetitive regions and detect a wider range of organisms in coinfections, while short-read platforms such as Illumina generally offer higher accuracy for distinguishing closely related species or low-abundance variants [38,41]. Furthermore, certain etiological agents pose added diagnostic hurdles. RNA viruses demand rapid turnaround for clinical relevance, where Nanopore’s real-time data generation is advantageous [29], whereas fungal pathogens often require deeper sequencing coverage to overcome low biomass, favoring Illumina’s higher throughput [32]. Thus, the interplay between specimen type, microbial diversity, and the suspected etiological spectrum is central to selecting the most appropriate sequencing platform for LRTI diagnostics.

### 4.1. Strengths and Limitations

This systematic review has several strengths. This is the only systematic review that has compared long-read with short-read metagenomics in LRTIs. Furthermore, a systematic search was undertaken in key databases to identify relevant studies. However, there are certain limitations as well that should be considered while interpreting the findings. The main limitation of our systematic review is that meta-analysis of the outcomes was not possible due to high variability in reported outcomes. For instance, only third studies reported sensitivity for both Illumina and Nanopore. Similarly, all other quantitative metrics were not reported consistently in all studies. The risk of bias in the included studies is high, with six studies showing patient selection bias, mainly manifested as not using the method of continuous inclusion of cases, which may lead to result bias; only three studies reported sensitive data from two platforms, with a small sample size, which may affect the stability of the results. Furthermore, there is a paucity of research on this topic currently, which limits the ability to draw conclusions regarding the approaches in LRTIs. Another limitation of the current study is that only a limited number of participants were included in the studies. Furthermore, we only included studies that directly compared platforms; we may have missed indirect comparisons or unpublished data. Furthermore, as the field of metagenomics is rapidly evolving, newer tools may change performance in the near future.

### 4.2. Implications for Practice and Research

The findings of the current systematic review can help clinicians and microbiologists to opt for the optimal platform when implementing mNGS for respiratory infections. In cases where speed is paramount, such as with critically ill patients, Nanopore can be used. Apart from having a shorter turnaround time, it also has the ability to detect multiple pathogen types, including DNA and RNA viruses, fungi, and atypical bacteria, which makes it versatile for undiagnosed pneumonia cases. On the other hand, for comprehensive profiling or outbreak sequencing, Illumina’s depth and accuracy may be preferable. In fact, a tiered approach involving Nanopore for initial rapid screening, then Illumina for confirmatory deep sequencing or longitudinal surveillance, can also be adopted. Regardless, both methods vastly outperform traditional culture in turnaround and yield. A major trend is adopting true real-time sequencing. Oxford Nanopore’s platforms (MinION, GridION) can begin analysis immediately as DNA passes through the pore. This enables rapid pathogen identification and antibiotic resistance detection at bedside. The review notes such promise; looking ahead, authors should emphasize integration of fully automated, bedside nanopore workflows. Recent commentary urges prospective trials of real-time metagenomics to assess impact on clinical outcomes. As the study by Gao et al. shows, combining mNGS with biopsy samples resulted in higher positive predictive value for identification of pathogens compared to traditional tests [42]. Emerging sequencing kits allow dozens of samples per run. For example, ONT’s 96-barcode kits and Illumina’s patterned flow cells with dual indexing can multiplex many respiratory samples. Future workflows will leverage this for outbreak surveillance or large studies. These findings also have implications for research. First, large-scale, multicenter trials are needed to validate these findings and to determine how platform choice affects patient outcomes. Most current studies are small or single-center. Pragmatic trials comparing Nanopore vs. Illumina-guided management would clarify real-world impact on antibiotic use and recovery. Furthermore, new research should try to set benchmark metrics that can allow replicability. Finally, as sequencing costs fall, detailed cost–benefit analyses will guide adoption. Future work should integrate clinical outcomes such as shortened ICU stays reduced broad-spectrum antibiotic use with economic models. Future studies should adopt continuous inclusion of cases and standardized reporting of various performance indicators.

## 5. Conclusions

This systematic review demonstrates that both long-read and short-read metagenomic sequencing platforms play crucial but distinct roles in profiling LRTIs. Illumina and other short-read platforms consistently demonstrated superior genome coverage, high read quality, and reliable stability. These attributes make short-read sequencing particularly suitable for confirmatory diagnostics, in-depth genome characterization, and antimicrobial resistance profiling. However, Illumina’s relatively slower turnaround time limits its utility in urgent clinical scenarios. Conversely, long-read platforms, particularly Nanopore, offered significant advantages in speed and pathogen breadth. Despite these strengths, Nanopore exhibited lower read accuracy and greater variability in specificity. Future studies should explore hybrid workflows combining rapid Nanopore-based screening with Illumina-based confirmation to optimize both speed and precision. To operationalize hybrid workflows, a two-tier pathway in which rapid Nanopore-based screening, with host depletion and rapid barcoding, delivers preliminary pathogen calls within hours, while parallel or reflex Illumina sequencing provides high-accuracy confirmation and in-depth genomic characterization within 24–72 h, should be adopted.

## Figures and Tables

**Figure 1 microorganisms-13-02366-f001:**
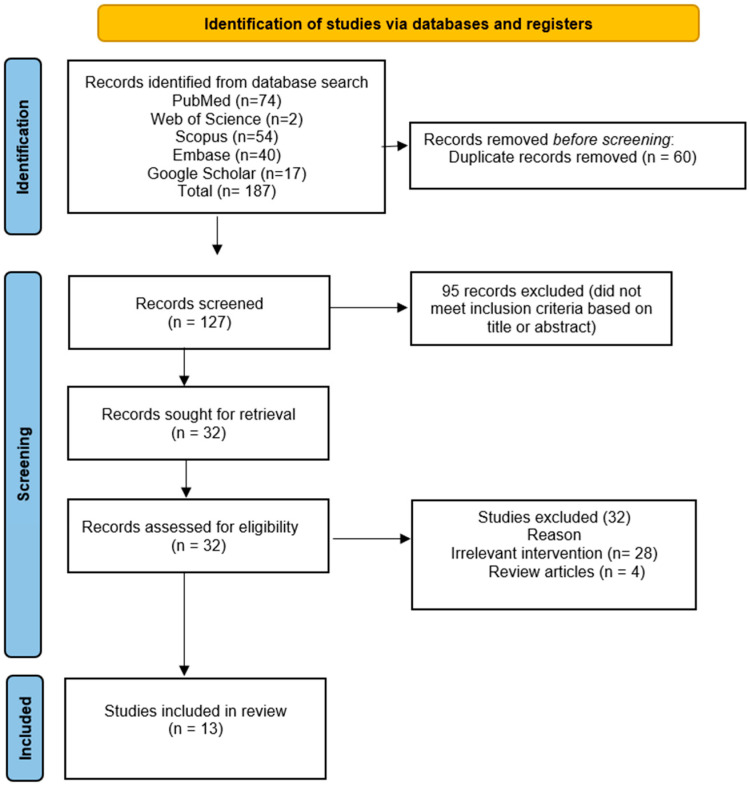
PRISMA flow diagram.

**Figure 2 microorganisms-13-02366-f002:**
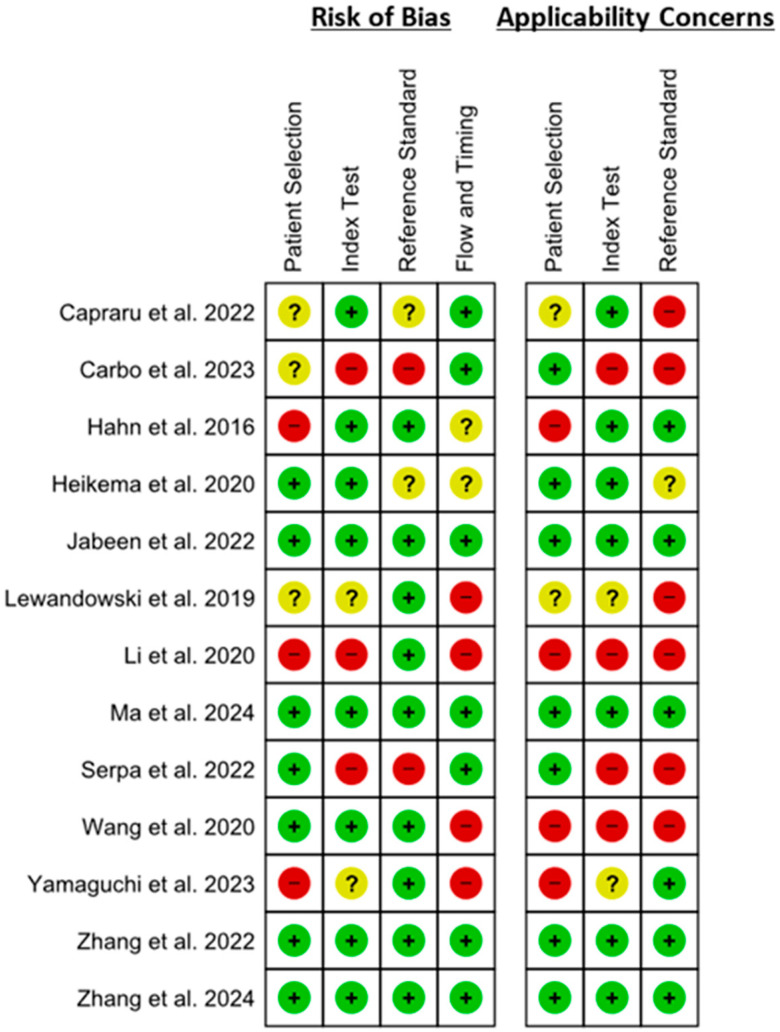
Risk of bias summary and applicability concern based on QUADAS-2. The figure presents the domain-level judgments of risk of bias (left panel) and applicability concerns (right panel) for each included study using the QUADAS-2 tool. Each row represents an individual study, and each column represents one of the QUADAS-2 domains. Symbols: Green (+) = low risk of bias/low concern for applicability. Yellow (?) = unclear risk of bias/unclear concern. Red (–) = high risk of bias/high concern [19,20,21,22,23,24,25,26,27,28,29,30,31].

**Figure 3 microorganisms-13-02366-f003:**
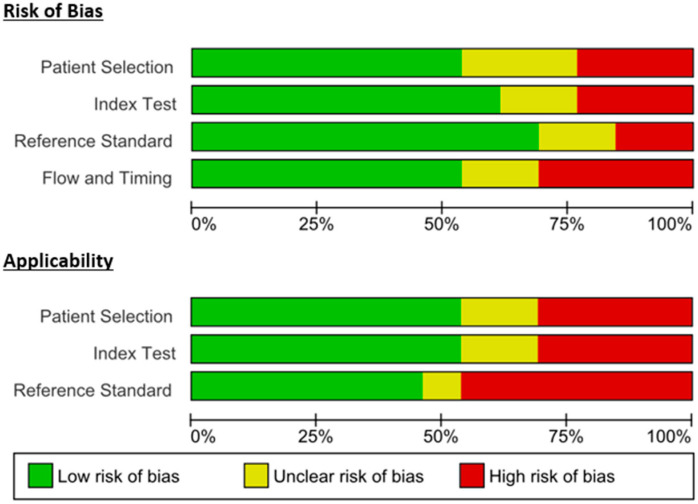
Risk of bias and applicability concern measured by QUADAS-2. Horizontal stacked bars show the proportion of studies judged to be at low (green), unclear (yellow), or high (red) risk of bias for each QUADAS-2 domain: patient selection, index test, reference standard, and flow and timing. The *x*-axis represents the percentage of the total number of studies (0–100%).

**Table 1 microorganisms-13-02366-t001:** Characteristics and diagnostic accuracy of included studies.

Authors	Year	Study Design	Participants	Age	Males	Comparator	Sensitivity	Specificity
Zhang et al. [21]	2022	Prospective	66	68 (58, 72)	63.6%	Illumina, Nanopore	Illumina: 46.7%, Nanopore: 40%	NA
Ma et al. [22]	2024	Retrospective	38	55 (36, 63)	63.20%	Illumina, Nanopore	Illumina: 80.6%, Nanopore: 93.5%	Illumina: 42.9%, Nanopore: 28.6%
Zhang et al. [26]	2024	Prospective	29	67 (65, 73.5)	NA	Illumina, Nanopore	Nanopore, 82.3%, Illumina, 88.2%	Nanopore, 75%, Illumina, 50%
Heikema et al. [27]	2020	Cross-sectional	10 adults and 49 children	NA	NA	Illumina, Nanopore	NA	NA
Carbo et al. [24]	2023	Retrospective	24	NA	NA	Illumina, Ion Torrent, and Nanopore	NA	NA
Capraru et al. [20]	2022	Cross-sectional analytical	103 samples	NA	NA	Ion torrent, Nanopore	NA	NA
Wang et al. [19]	2020	Case study	63-year-old male	-	-	Nanopore and BGISEQ-500	-	-
Lewandowski et al. [28]	2019	Methodological validation study	50 samples	NA	NA	Illumina and nanopore	Nanopore: 83%, Illumina: NA	Nanopore: 100%, Illumina: NA
Yamaguchi et al. [29]	2023	Prospective	31 samples	NA	NA	Illumina and nanopore	NA	NA
Hahn et al. [25]	2016	Cross-sectional	12 samples	NA	NA	PacBio, Illumina	NA	NA
Jabeen et al. [30]	2022	Prospective	23	67 (10)	57%	Illumina, Nanopore	NA	NA
Serpa et al. [31]	2022	Retrospective	88	NA	NA	Illumina, Nanopore	Illumina: Gram-positive: 70%, Gram-negative: 100%, Nanopore: 100%	Illumina: Gram-positive: 95%, Gram-negative: 64%, Nanopore: NA
Li et al. [23]	2020	Cross-sectional diagnostic accuracy study	29 clinical SARS-CoV-2 specimens	NA	NA	Illumina, Nanopore	NA	NA

**Table 2 microorganisms-13-02366-t002:** Performance characteristics and outcomes of long-read vs. short-read platforms.

Authors	Platform	Concordance/Agreement (%)	Positivity Rate	Turn-Around Time	Genome Coverage (%)	Read Metrics	Other Outcomes	Conclusion
Zhang et al. [21]	Illumina	56.1%	NA	20 (19–21)	NA	NA	AUC Bacteria: 0.73, Fungi: 0.73	Nanopore detected more taxa overall than Illumina.
	Nanopore	57.6%	NA	14 (11–15)	NA	NA	AUC Bacteria: 0.60, Fungi: 0.81	
Ma et al. [22]	Illumina	63.9%	Bacteria: 71.4%, Fungi: 50%	NA	NA	NA	61.1% detected (with antibiotics) and 46.2% detected (without antibiotics)	Nanopore sequencing showed higher sensitivity and better concordance than Illumina, particularly for detecting *Mycobacterium*.
	Nanopore	83.3%	Bacteria: 78.6%, Fungi: 62.5%	NA	NA	NA	77.8% detected (with antibiotics) and 76.9% detected (without antibiotics)	
Zhang et al. [26]	Illumina	NA	51.7%	24	NA	NA	NA	Nanopore exhibited relatively better consistency.
	Nanopore	NA	48.3%	8	NA	NA	Required shorter time	
Heikema et al. [27]	Illumina	69.1% with nanopore	91%	NA	NA	131,024	ISI: 2.7, Mean genera detected (≥1%): 4.4	Both comparable but nanopore is not that effective with genus *Corynebacterium*
	Nanopore	-	78%	NA	NA	21,907	ISI: 2.2, Genera detected: 4.5	
Carbo et al. [24]	Illumina	NA	NA	3 days	99.8%	Depth: 860	NA	Illumina has higher accuracy but longer time.
	Nanopore	NA	NA	<24 h	81.2%	Depth: >2000	NA	
Capraru et al. [20]	Ion torrent	Clade: 90.90%	NA	NA	NA	190 base pair	NA	Nanopore is faster with deeper coverage; Ion Torrent higher alignment rates
	Nanopore		NA	NA	>250×	519.17 bases	NA	
Wang et al. [19]	BGISEQ-500	100%	-	NA	100%	129,512,318	NA	Both rapidly and reliably identified the causative pathogen.
	Nanopore	100%	-	12.14 h	45%	34,831	NA	
Lewandowski et al. [28]	Illumina	100% with nanopore	NA	NA	26.6%	NA	NA	Nanopore is comparable to Illumina in sequencing influenza viruses.
	Nanopore	-	NA	NA	≥99.3% per segment	3.8 × 105 reads	Limit of Detection: 10^2^–10^3^ copies/mL	
Yamaguchi et al. [29]	Illumina	Reference	NA	NA	NA	2,155,152, 264,467,762 bases	NA	In a comparison of 7 BALF samples, nanopore sequencing detected the same RNA viruses as Illumina.
	Nanopore	71.4%	41.7%	NA	81.38%	220,600, 699,203,556 bases	NA	
Hahn et al. [25]	Illumina	NA	49.4%	NA	NA	479,220 reads Per-sample	MiSeq sequencing of the 16S rRNA V4 region provided higher alpha-diversity estimates	PacBio identified *Burkholderia* while MiSeq detected more *Escherichia*.
	PacBio	NA	99.3%	NA	NA	122,526 reads Per-sample		
Jabeen et al. [30]	Illumina	NA	NA	NA	NA	172 base pair	Nanopore sequencing achieved near-complete genome coverage and depth at all read–depth thresholds compared with MiSeq	
	Nanopore	NA	NA	NA	24.2–94.2	2013 base pair		
Serpa et al. [31]	Illumina	100% of AMR loci identified by Illumina	NA	NA	NA	6.9 × 10^7^ reads per sample	NA	Illumina and nanopore has similar sensitivity
	Nanopore	81% of culture-confirmed bacterial pathogens	NA	NA	NA	1.19 × 10^6^ total reads per sample	NA	
Li et al. [23]	Illumina	NA	NA	NA	NA	NA	NA	Nanopore detected whole genomes from samples diluted up to 100,000× (undetectable by qRT-PCR), with ≥97.6% completeness at >250× depth
	Nanopore	100% with Illumina	NA	NA	98.08–100%	NA	NA	

## Data Availability

No new data were created or analyzed in this study.

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
