# Peer review of "Comparative Meta-Analysis of Long-Read and Short-Read Sequencing for Metagenomic Profiling of the Lower Respiratory Tract Infections"

_microorganisms, 2025, doi:10.3390/microorganisms13102366_

Round 1
Reviewer 1 Report
Comments and Suggestions for Authors
This review evaluates the comparative performance of long-read and short-read metagenomic sequencing for the diagnosis and clinical application in lower respiratory tract infections (LRTI). The manuscript currently reads more as a general comparison of the studies between specific platforms (e.g., Illumina vs. Nanopore). It is unclear how this review provides novel insights specific for LRTI beyond a compilation of existing results.
Comments for authors to address
Major
- While the manuscript summarizes findings from individual studies, it would benefit from a stronger synthesis of results across the literature. The authors should consider going beyond reporting each study separately and instead analyze the collective evidence to highlight consistent patterns and important differences. e.g., The section describing the QUADAS-2 results is difficult to follow, as it lists the number of studies in each risk category without synthesizing the overall patterns.
- The manuscript could better clarify how LRTI presents unique challenges to metagenomics compared with other infection sites, and if these differences are important to help decide on the metagenomic platform. Do specimens affect the platforms performance? How about etiological agents (e.g., virus or fungi) or coinfections?
Minor
- Figures 2 and 3 lack legends, making them difficult to interpret. These should be added for clarity.
Author Response
Comments1: While the manuscript summarizes findings from individual studies, it would benefit from a stronger synthesis of results across the literature. The authors should consider going beyond reporting each study separately and instead analyze the collective evidence to highlight consistent patterns and important differences. e.g., The section describing the QUADAS-2 results is difficult to follow, as it lists the number of studies in each risk category without synthesizing the overall patterns.
Response: Unfortunately, the overall summary of the results was not possible because all outcomes were not reported across studies. However, we did provide a range of outcomes. We have mentioned this in the limitation section as well. We have mentioned overall pattern in QUADAS-2 as well.
Comments 2: The manuscript could better clarify how LRTI presents unique challenges to metagenomics compared with other infection sites, and if these differences are important to help decide on the metagenomic platform. Do specimens affect the platforms performance? How about etiological agents (e.g., virus or fungi) or coinfections?
Response: We have clarified this at the end of discussion.
Comments 3: Figures 2 and 3 lack legends, making them difficult to interpret. These should be added for clarity.
Response: We have added the figure legend.
Reviewer 2 Report
Comments and Suggestions for Authors
This review compares the efficacy of long read and short read mNGS in diagnosing lower respiratory tract infections. The results showed that both have comparable overall sensitivity, but each has its own advantages: Illumina has high accuracy and Nanopore has fast speed. However, there is a high risk of bias in the research, and the conclusions need to be interpreted with caution. This study provides evidence-based support for the clinical selection of mNGS platforms and has certain clinical reference value.
My suggestions and comments are as follows:
1, Abstract section: The conclusion only points out the advantages and applicable scenarios of two sequencing platforms, without mentioning the limitations of this review that affect the credibility of the conclusion. A brief supplement should be provided.
2, Introduction section: Although it mentions the limitations of traditional culture methods and the advantages of metagenomic sequencing (mNGS), it does not specifically explain the urgent clinical need for rapid and accurate pathogen detection in LRTIs diagnosis and the direct correlation between long-read and short-read sequencing technology. It is recommended to supplement.
3, Result section: The average sensitivity and specificity ranges of Illumina and Nanopore were mentioned, but sensitivity and specificity data were not presented separately by pathogen type (such as bacteria, fungi, viruses, and mycobacteria). It is recommended to supplement the stratified analysis results to more accurately reflect the performance differences between the two platforms.
4, Discussion section: Two applicable scenarios for the two platforms were mentioned, but the potential challenges in actual clinical applications were not analyzed. It is suggested to supplement the discussion to make it more relevant to clinical practice.
5, Discussion section: The current limitation analysis is relatively general and needs further refinement, such as "the risk of bias in the included studies is high, with six studies showing patient selection bias, mainly manifested as not using the method of continuous inclusion of cases, which may lead to result bias; only three studies reported sensitive data from two platforms, with a small sample size, which may affect the stability of the results", and specific improvement directions should be proposed for the limitations (such as future studies should adopt continuous inclusion of cases and standardized reporting of various performance indicators).
6, Conclusion section: It mentions "exploring hybrid workflows" but does not specify the specific implementation path of hybrid workflows. It is recommended to supplement actionable research directions to provide clearer guidance for future research.
Author Response
Comments 1: Abstract section: The conclusion only points out the advantages and applicable scenarios of two sequencing platforms, without mentioning the limitations of this review that affect the credibility of the conclusion. A brief supplement should be provided.
Response: We made modifications in the manuscript accordingly.
Comments 2: Introduction section: Although it mentions the limitations of traditional culture methods and the advantages of metagenomic sequencing (mNGS), it does not specifically explain the urgent clinical need for rapid and accurate pathogen detection in LRTIs diagnosis and the direct correlation between long-read and short-read sequencing technology. It is recommended to supplement.
Response: We have added an urgent need for the detection of pathogens in LRTIs in the introduction.
Comments 3: Result section: The average sensitivity and specificity ranges of Illumina and Nanopore were mentioned, but sensitivity and specificity data were not presented separately by pathogen type (such as bacteria, fungi, viruses, and mycobacteria). It is recommended to supplement the stratified analysis results to more accurately reflect the performance differences between the two platforms.
Response: Unfortunately, this was not possible because this outcome was reported only by 1-2 studies. Therefore, it was not possible to analyze performance differences between platforms.
Comments 4: Discussion section: Two applicable scenarios for the two platforms were mentioned, but the potential challenges in actual clinical applications were not analyzed. It is suggested to supplement the discussion to make it more relevant to clinical practice.
Response: We have mentioned potential challenges regarding the clinical application of these platforms.
Comments 5: Discussion section: The current limitation analysis is relatively general and needs further refinement, such as "the risk of bias in the included studies is high, with six studies showing patient selection bias, mainly manifested as not using the method of continuous inclusion of cases, which may lead to result bias; only three studies reported sensitive data from two platforms, with a small sample size, which may affect the stability of the results", and specific improvement directions should be proposed for the limitations (such as future studies should adopt continuous inclusion of cases and standardized reporting of various performance indicators).
Response: We have further mentioned these limitations in the manuscript.
Comments 6: Conclusion section: It mentions "exploring hybrid workflows" but does not specify the specific implementation path of hybrid workflows. It is recommended to supplement actionable research directions to provide clearer guidance for future research.
Response: We have further explained hybrid workflows in conclusion.
Round 2
Reviewer 1 Report
Comments and Suggestions for Authors
Authors have addressed the comments.
Reviewer 2 Report
Comments and Suggestions for Authors
The author has made revisions according to the reviewer's comments, and the quality has improved. The reviewer has no further suggestions or comments.